# Solving the Tree Containment Problem Using Graph Neural Networks

**Arkadiy Dushatskiy**                                                    *Arkadiy.Dushatskiy@cwi.nl*
*Centrum Wiskunde & Informatica*
*Amsterdam, the Netherlands*

**Esther Julien**                                                        *E.A.T.Julien@tudelft.nl*
*Delft University of Technology*
*Delft, the Netherlands*

**Leen Stougie**                                                         *Leen.Stougie@cwi.nl*
*Centrum Wiskunde & Informatica*
*Vrije Universiteit Amsterdam*
*Amsterdam, the Netherlands*

**Leo van Iersel**                                                       *L.J.J.vanIersel@tudelft.nl*
*Delft University of Technology*
*Delft, the Netherlands*

**Reviewed on OpenReview:** *https://openreview.net/forum?id=nK5MazeIpn*

## Abstract

TREE CONTAINMENT is a fundamental problem in phylogenetics useful for verifying a proposed phylogenetic network, representing the evolutionary history of certain species. TREE CONTAINMENT asks whether the given phylogenetic tree (for instance, constructed from a DNA fragment showing tree-like evolution) is contained in the given phylogenetic network. In the general case, this is an NP-complete problem. We propose to solve it approximately using Graph Neural Networks. In particular, we propose to combine the given network and the tree and apply a Graph Neural Network to this network-tree graph. This way, we achieve the capability of solving the tree containment instances representing a larger number of species than the instances contained in the training dataset (i.e., our algorithm has the inductive learning ability). Our algorithm demonstrates an accuracy of over 95% in solving the tree containment problem on instances with up to 100 leaves.

## 1 Introduction

Evolutionary trajectories of biological species can be described using directed acyclic graphs called *phylogenetic networks*, which allow the representation of common evolutionary events such as lateral gene transfer and hybridization (Bapteste et al., 2013). Leaves of phylogenetic networks are labeled with the names of studied species. In many cases, some parts of the DNA structure evolve in graphs that are trees (Janssen & Murakami, 2021). A valid phylogenetic network representing the evolution of certain species should be consistent with the previously built trees, namely, it should contain the trees which represent the evolutionary history of some pieces of DNA of the considered species. Consequently, a mathematical problem arises called TREE CONTAINMENT: the task is to determine whether a given phylogenetic tree is contained by a given phylogenetic network.

In general, the tree containment problem is *NP-complete* (Kanj et al., 2008). However, for some classes of networks (with certain topological restrictions), polynomial-time algorithms (in some cases, even linear) were proposed, e.g., by Janssen & Murakami (2020), Gambette et al. (2018), and Janssen & Murakami (2021).

An algorithm that solves the tree containment problem should make use of the labels of the leaves of both the given phylogenetic network and the tree, as the containment property depends on them (the formal definition is provided in Section 3.2). This naturally poses a challenge in using machine learning approaches to solving it. It is especially difficult to handle situations when the number of leaves in the test instances can be larger than in the training ones. In this paper, we propose to tackle this challenge and develop an approach to approximately solving TREE CONTAINMENT, including instances for which no polynomial-time algorithms are known. The main contributions of this paper can be formulated as follows:

1. We propose an approximate solution to the NP-complete tree containment problem using Graph Neural Networks. To the best of our knowledge, this is the first time a machine learning approach has been proposed to address this problem. Our approach can be applied to evaluate the quality of a constructed phylogenetic network: it can be used to efficiently and with high accuracy estimate whether trees (e.g., gene trees) are contained in the built network.

2. Our proposed approach demonstrates a generalization ability: when trained on smaller instances (phylogenetic networks and trees with a smaller number of studied species), it achieves high accuracy (on average, over 95%) on larger instances not included in the training dataset. This inductive learning ability is achieved through a specific technique: combining the given phylogenetic network and tree into a single graph while respecting the labels of the leaves, which is crucial for solving the tree containment problem.

3. This work shows the potential of using GNNs for solving phylogenetic problems. Potentially, ideas from the proposed approach to solving tree containment, might be applied to solving other phylogenetic problems as well.

## 2 Related work

### Graph Neural Networks

Graph Neural Networks (GNNs) are a broad class of deep neural networks, which can be applied to solve graph problems (e.g., node and edge classification, graph classification) and can learn meaningful representations of nodes, edges, and entire graphs. Message Passing Neural Networks (MPNN) (Gilmer et al., 2017) is a general framework of neural networks for graphs. In each layer of an MPNN, each node receives messages containing information about its neighbors, then those messages are aggregated and combined with the node information itself. Examples of popular algorithms which use the message-passing paradigm are Graph Convolutional Networks (GCN) (Kipf & Welling, 2017), Graph Attention Networks (GAT) (Veličković et al., 2018), GraphSAGE (Hamilton et al., 2017), and GIN (Xu et al., 2019). The comprehensive overview of deep learning methods for graphs, including the recently proposed and non-MPNN ones can be found in Khoshraftar & An (2024).

### GNNs for directed graphs

The phylogenetic networks and trees in the tree containment problem are naturally directed graphs. Most of the GNNs work with undirected graphs. However, many real-world graphs are directed, and therefore several approaches have been proposed to adapt GNNs to directed graphs, e.g., (Zhang et al., 2021b). Recently a more general approach to using GNNs for directed graphs was proposed in Rossi et al. (2023). This approach called *Dir-GNN* performs message passing in two directions (basically transforming the graph to an undirected one), but considers each direction of message passing separately with its own learnable weights for each. This ultimately results in GNNs with more expressive power (Rossi et al., 2023). Dir-GNN was shown to be a general approach applicable to different GNNs such as GCN (Kipf & Welling, 2017), GAT (Veličković et al., 2018), and GraphSAGE (Hamilton et al., 2017).

**The tree containment problem**

Different approaches have been proposed to solve TREE CONTAINMENT in polynomial time for some specific classes of networks (that have certain restrictions on the structure). Examples of such classes are binary near-stable networks (Gambette et al., 2018) and binary tree-child networks (Gunawan, 2018). In Janssen & Murakami (2020), a more general version of the tree containment problem was considered (a network in network containment instead of a tree in network containment), and a linear-time algorithm was proposed to solve this problem for semi-binary tree-child networks. In the follow-up work (Janssen & Murakami, 2021), a linear algorithm for tree (network) containment was also proposed for the so-called cherry picking networks and rigorous proofs of algorithm complexity were provided. A different approach was proposed in van Iersel et al. (2023), and Huijsman (2023): the main idea behind it is to design an algorithm that has an exponential time complexity in terms of some input parameter, which results in a polynomial complexity when this parameter is constant (fixed-parameter tractability). Specifically, the treewidth (measuring the tree-likeness of a network) was considered as such a parameter.

**Solving hard problems with Graph Neural Networks**

GNNs have been applied to solving challenging, in many cases, NP-complete and NP-hard combinatorial problems such as Traveling Salesman, Maximum Cut, Minimum Vertex Cover (Barrett et al., 2020; Cappart et al., 2023; Li et al., 2018; Kool et al., 2019). Usually, in such approaches, it is proposed to use GNNs as part of the reinforcement learning algorithms: they can extract useful features from a problem instance and a partial solution, and help in building a high-quality solution further.

GNNs were also applied to solving a similar type of combinatorial problem to the considered tree containment problem: graph matching and (sub-)Graph Edit Distance (GED) calculation problems. There is a rather broad collection of works in which GNNs are applied to (sub)graph matching problems. One common approach (Bai et al., 2019; 2020; Lou et al., 2020) is to learn meaningful embeddings of two graphs and then use them as inputs to the predictor, which outputs the distance between the graphs. An alternative approach that is used, for instance, in Doan et al. (2021); Bai et al. (2020), is to have an explicit algorithm of matching the nodes of one graph with the corresponding nodes of another (using the learned node-wise embeddings). In Zhang et al. (2021c) it was proposed to use the hypergraph concept to further improve the algorithm performance. Recently, Ranjan et al. (2022) proposed essentially a simpler approach to (sub)graph edit distance learning that demonstrated state-of-the-art performance outperforming previous works such as Lou et al. (2020). In contrast to previous works, it does not construct an explicit node-to-node similarity matrix, but instead just translates the distance between embeddings of two graphs into their distance function. The embeddings of two graphs are obtained using Siamese GNNs similar to Xu et al. (2019).

We note that the above-mentioned approaches to the graph matching and GED problems are not directly applicable to TREE CONTAINMENT as, in contrast to the standard graph matching formulation, it has some specific rules on node mapping, i.e., leaves mapping should respect their labels (more details are provided in Section 3.2).

To the best of our knowledge, applying GNNs to solving phylogenetic problems has not been widely explored yet. The existing works in this field (Zhang, 2022; Mimori & Hamada, 2023) mainly focus on the learning node representations in the scope of phylogenetic inference, and we note that the phylogenetic problem considered in this work is different from that.

## 3 Preliminaries

### 3.1 Phylogenetic networks

A rooted phylogenetic network constructed for a set of taxa $X$ is a directed acyclic graph (DAG) with leaves labelled from the set of labels $x \in X$ and with some specific restrictions on the in- and out-degrees of the nodes: the network contains a single root (in-degree zero); all other nodes are either a leaf (in-degree one, out-degree zero), a tree node (in-degree one, out-degree greater than one), or a reticulation node (in-degree greater than one, out-degree one). In this paper, we consider binary phylogenetic networks, meaning that

in- and out-degrees of all nodes are at most two. A phylogenetic tree is a special case of a phylogenetic network: it is a phylogenetic network that contains no reticulation nodes.

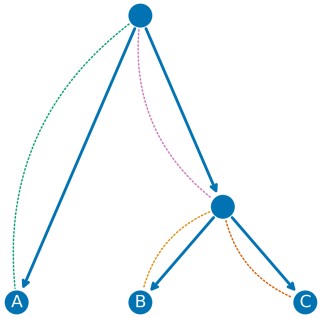

(a) A phylogenetic tree.

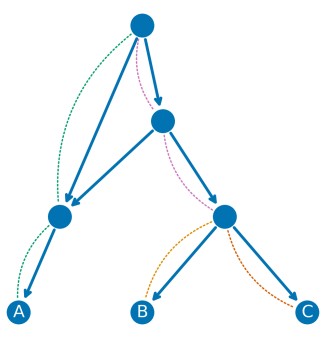

(b) A phylogenetic network that contains the given tree (a).

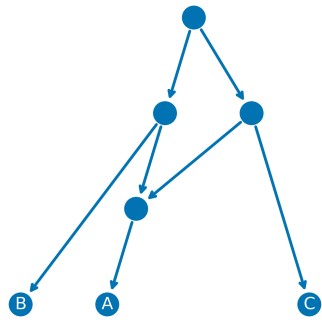

(c) A phylogenetic network that does not contain the given tree (a). Note: the leaves' labels have a different left-to-right order.

Figure 1: Examples of the tree containment problem instances: a tree (a) and two networks, one (b) does contain the tree, and one (c) does not. Leaves are labeled with letters (representing a set of taxa). The mapping between the edges of the tree (a) and the paths of the network (b) (as explained in Section 3) is depicted with the dashed curves with of corresponding colors. For the second network (c) no such mapping exists.

## 3.2 Tree containment problem

The tree $T$ (in the general case, a network), $T = (V', E', X')$) is *contained* (alternative terms *embedded* and *displayed* can also be used) in network $N = (V, E, X)$ if there exists an injective mapping of the nodes of $T$ to the nodes of $N$: $V' \mapsto V$ and edges of $T$ to node-disjoint paths in $N$, such that the leaves' labels are preserved and the mapping of the edges respects the mapping of the nodes. In other words, $T$ is contained in $N$, if N has a subgraph $T'$, such that it can be obtained by replacing some edges of $T$ by (directed) paths. Examples of cases where a tree is contained and is not contained in a network are shown in Figure 1. In this paper, we consider the problem of determining whether a phylogenetic tree on taxa $X$ is contained in a binary phylogenetic network on the same taxa $X$: given a tree $T$ and a network $N$, the task is to give an answer on whether $T$ is contained in $N$ or not:

---

NETWORK CONTAINMENT
**Input**: binary phylogenetic network $N$ and tree $T$, both have the same set of leaf labels $X$
**Question**: is $T$ displayed in $N$?

---

In the general case, when the network does not belong to one of the specific classes, e.g., tree-child networks (Pons & Batle, 2021) (such networks that each non-leaf node has a child that is either a tree vertex or a leaf), this is a NP-complete problem (Kanj et al., 2008).

# 4 Solving the tree containment problem using a GNN

## 4.1 The summary of our approach

The primary challenge when applying a GNN to solve the tree containment problem involves ensuring its awareness of the labels on the leaves while retaining the ability to generalize to instances beyond the training set (inductive learning ability). For example, employing the labels of one-hot encoded leaves as node features, (as, e.g., in Zhang et al. (2021a)), does not enable inductive learning ability. Completely ignoring the labels of the leaves is a naive approach that, generally, can not be expected to work well, because the answer to

the tree containment problem might differ for two pairs $(N, T)$ and $(N', T')$ where $N$ and $N'$ have identical topologies but different leaf labeling (similarly for $T$ and $T'$).

Our proposed approach to address the aforementioned challenge of applying GNNs is as follows: we combine a network and a tree into a larger *network-tree graph*, ensuring shared corresponding leaves. This graph is also called the *display graph* in the phylogenetic literature (Bryant & Lagergren, 2006; van Iersel et al., 2023). This process is illustrated in Figure 2 (left side). With this technique, the labels of the leaves are taken into account in the network-tree graph construction as they should be considered to correctly solve the tree containment. However, we do not use the *specific* leaves' labels (for instance, as node features), therefore, this approach, potentially, can work with the labels which have not been seen in the training data. In other words, it holds the potential to enable inductive learning ability.

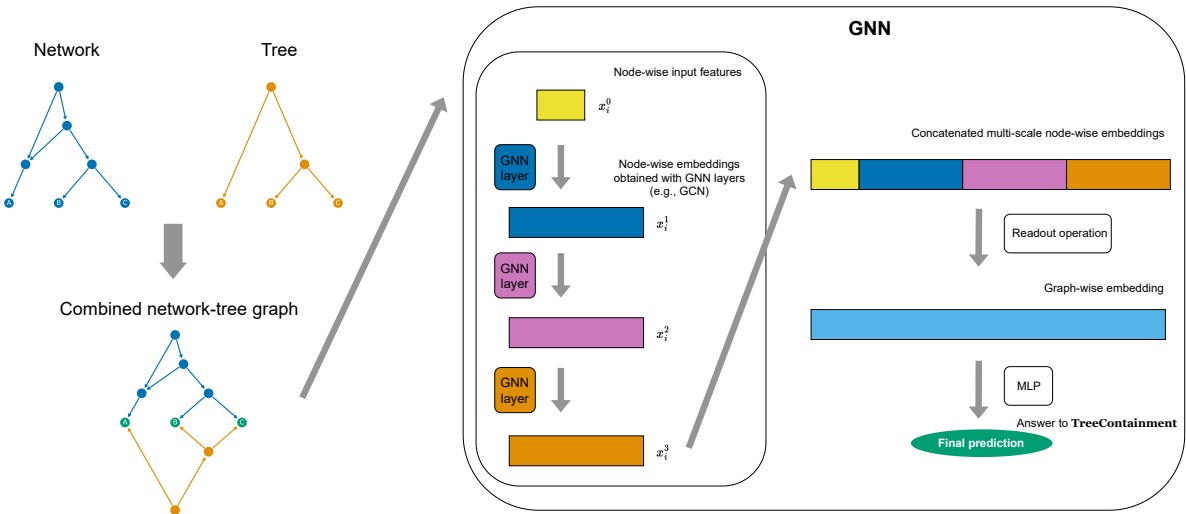

Figure 2: The schematic illustration of our approach (*Combine-GNN*). To solve a tree containment problem for the given network and tree, we start by combining them in a single graph such that the leaves are shared, respecting their labels. Then, a GNN is applied. We note that the input node features do not contain the leaves' labels. The GNN consists of multiple layers, then the obtained embeddings are concatenated, and a node aggregation (graph readout) is applied. Finally, an MLP is used to produce the final prediction.

After obtaining a network-tree graph, we apply a GNN to it. The architecture of the used GNN is depicted in Figure 2 (right side). The goal of the GNN is to extract the necessary information from the graph structure and produce the answer to the tree containment problem. First, we use a multi-layer message-passing GNN to obtain meaningful embeddings of the nodes (a comprehensive survey of various GNNs can be found, for instance, in Hamilton (2020)).

As the considered phylogenetic networks and trees are directed graphs, simply transforming them into undirected ones before passing to a GNN might lead to the loss of some information. On the other hand, if the graphs are kept directed, without special adaptations, message-passing GNNs cannot be efficiently applied to our problem as, for instance, leaves, would not pass any messages (their out-degree is zero) to other nodes, and this also leads to the loss of information. To take into account the directed nature of the graphs in the tree containment problem, we use the Dir-GNN approach (Rossi et al., 2023). It allows bi-directional message-passing, but has separate learnable weights of AGGREGATE operations in each direction. The message-passing part of the GNN in our approach is therefore as follows:

$$m_{i,\leftarrow}^k \leftarrow \text{AGGREGATE}_{i,\leftarrow}^k \left( \{ (x_j^{k-1}, x_i^{k-1}) : (j, i) \in E \} \right)$$
$$m_{i,\rightarrow}^k \leftarrow \text{AGGREGATE}_{i,\rightarrow}^k \left( \{ (x_j^{k-1}, x_i^{k-1}) : (i, j) \in E \} \right)$$
$$x_i^k \leftarrow \text{COMBINE}^k (x_i^{k-1}, m_{i,\leftarrow}^k, m_{i,\rightarrow}^k)$$

where $x_i^k$ is a node embedding (associated feature vector) of node $i$ at layer $k$, $m_{i,\leftarrow}^k$ and $m_{i,\rightarrow}^k$ are passed messages in two directions. AGGREGATE and COMBINE are crucial operations with usually learnable parameters (weights), and their specific implementations vary in different GNNs. For instance, in GCN (Kipf & Welling, 2017), they are implemented as averaging with a consequent linear operation, usually followed by applying a non-linear function (e.g., ReLU).

After obtaining node-wise embeddings, we gather and concatenate the embeddings from different layers of the network, similar to the Jumping Knowledge GNN (Xu et al., 2018) and Ranjan et al. (2022):

$$x_i^{final} \leftarrow \text{CONCATENATE}([x_i^0, x_i^1 \dots, x_i^L]),$$

where $L$ is the number of layers, $x_i^0$ are the features passed to the first layer. Here, we pass the normalized original node features of the graph, followed by a linear projection (this projection might, in principle, be omitted). With multiple GNN layers, we obtain a multi-scale representation of the graph nodes. Subsequently, these concatenated embeddings are utilized to obtain the whole graph representation using one of the readout operations (e.g., an element-wise average pooling). Finally, a Multilayer Perceptron (MLP) is employed to make the final prediction:

$$G \leftarrow \text{READOUT}(\{x_1^{final}, \dots, x_N^{final}\})$$
$$output \leftarrow \text{MLP}(G)$$

We call the proposed approach to solving the tree containment problem ***Combine-GNN***. This name reflects the two main ideas behind it: combining a network and a tree in a single graph, and applying a GNN to it. The code and used tree containment instances are available at https://github.com/ArkadiyD/PhyloGNN.

### 4.2 Node features

The considered phylogenetic graphs naturally do not have node features, except for the leaves (they have the labels). However, we generate some features as a preprocessing step in order to provide additional helpful information to a GNN for solving the tree containment problem. We use two types of features for the nodes of the obtained network-tree graphs: 1) In- and out-degrees of the nodes; 2) Node origin: for each node of the network-tree graph, we specify whether the node was a leaf, was part of the network or of the tree. This information is provided in the form of one-hot encoding. Without explicitly providing this information, the GNN can try to derive it implicitly, but this might complicate the learning process.

### 4.3 Time complexity

As analyzed, for instance, in Kipf & Welling (2017), the time complexity of a GNN consisting of multiple GCN layers is $\mathcal{O}(|E|)$ (assuming the number of layers and node features are constants). In our case, this means $\mathcal{O}(|E_{network}| + |E_{tree}|)$, which is equal to $\mathcal{O}(|E_{network}| + |E_{network}|) = \mathcal{O}(|E_{network}|)$ because if a tree is larger than the network than it is definitely not contained in it, and we do not have to run the algorithm to determine the containment. $\mathcal{O}(|E_{network}|)$ is, in turn, equal to $\mathcal{O}(N)$ ($N$ is the number of nodes in the network) due to the sparse nature of binary phylogenetic networks. The usage of Dir-GNN and data preprocessing (obtaining the network-tree graph) do not change the time complexity of Combine-GNN. The data processing step (including the network-tree graph creation) has $\mathcal{O}(N)$ complexity: creating node features (specifically, indicators of the node origin, see Section 4.2) has linear complexity as the features need to be created for each node.

## 5 Experiments

### 5.1 Data

Our datasets consist of network-tree pairs ($N$, $T$) along with corresponding labels indicating tree containment (label "one" indicating the network contains the tree, label "zero" indicating no containment).

**Synthetic data**

The data generation process is similar to the one used in Bernardini et al. (2023). First, we generate a network using a Lateral Gene Transfer (LGT) generator from Pons et al. (2019) and randomly select a tree subgraph of it. This way, we obtain a positive example (the tree is contained in the network). To obtain a negative example (a tree is not contained in the network), we perform a number of so-called *tail moves* (Janssen, 2021). A tail move is defined on two edges $e = (u, v)$ and $f$: it simply moves the node $u$ to the edge $f$, with respect to the corresponding edges. A valid tail move guarantees that all necessary properties of the phylogenetic network (listed in Section 3) are kept intact. Furthermore, a tail move does not change the main graph properties (number of nodes, number of edges, etc.). After performing a number of tail moves the tree containment is checked using the exact algorithm called BOTCH (van Iersel et al., 2023; Huijsman, 2023).

Intuitively, determining whether a tree is contained within a network becomes more challenging when the network under consideration resembles another network that contains the specified tree. Thus, we can adjust the difficulty of the problem by varying the number of tail moves performed. A higher number of tail moves means greater modifications to the network, resulting in a network more dissimilar to the original, which contains the given tree. We create rather challenging instances by using the number of tail moves sampled uniformly at random from the integers in the range $[1; 5]$.

We vary the number of leaves in the generated datasets. We consider two different scenarios: 1) *Transductive learning*: the number of leaves in the validation/test instances does not exceed the number of leaves in the training data; 2) *Inductive learning*: the number of leaves in the validation/test instances is in the interval $(L_{train}; 2L_{train}]$, while the number of leaves in the training set is in $[L_{min}, L_{train}]$. The minimum number of leaves $L_{min}$ in all instances is 5, and the maximum number considered is 100. We sample the parameter $\alpha$ of the LGT generator uniformly randomly from $[0.1; 0.2]$. The summary statistics of the generated graphs are provided in Appendix, Figure 10.

In each setup (defined by the number of leaves in the training and validation/test instances), we have 10000 training samples (network-tree pairs), 1000 samples used as validation dataset, and 1000 for the test dataset.

**Real-world data**

For the the real-world experiments, we use the dataset of gene trees from bacterial and archaeal genomes (proposed in Beiko (2011), binarized in van Iersel et al. (2022)). For each set of input trees, we construct a network with the minimized number of reticulations such that it contains all given trees using the TrivialRand algorithm from Bernardini et al. (2023). Then, to obtain negative examples, we perform tail moves on the network similar to the synthetic datasets construction. As obtaining the correct tree containment labels using BOTCH is becoming extremely time consuming for large networks (that have large number of leaves and/or high number of reticulations), we limit the considered instances to maximum 50 leaves and 5 input trees. As we have limited number of real-world instances (less than 300 network-tree pairs for each value of $L \in \{10, 20, 30, 40, 50\}$, we use them only as validation and test data while using the synthetic LGT data for training. We also do not specifically tune any hyperparameters on the real-world data. For this experiment, we sample the parameter $\alpha$ of the LGT generator uniformly randomly from $[0.3; 0.5]$ (to increase the number of reticulation nodes in the networks). The number of leaves in the train instances is from $[5; 50]$ for all test instances.

## 5.2  Baselines

The first baseline we consider is a naive approach that disregards the labels of the leaves. Instead, it attempts to predict the tree containment property based on manually generated features from both the network and the tree, such as the average distance from the root to the leaves. The complete list of used features is listed in Appendix, Table 2. We train an XGBoost model (Chen & Guestrin, 2016) using these features.

Next, we consider a Siamese GNN (similar to the GREED algorithm (Ranjan et al., 2022)) applied to the tree containment problem. We make it aware of the labels of the leaves by using one-hot encoding of the labels as the node features. This approach does not enable the inductive learning ability, but could potentially

work well in the transductive learning setting. Moreover, in theory, it can use the automatically extracted features from the network and the tree (in contrast to the manually constructed features used in the first considered baseline).

### 5.3  GNN architecture and training hyperparameters tuning

We train GNNs using the AdamW optimizer (Loshchilov & Hutter, 2017) with the cosine annealing learning rate schedule, number of training steps is 5000; batch size is 200 (similar parameters were used in Rossi et al. (2023)).

We performed a grid search to find the best performing architecture and the corresponding training hyperparameters for Combine-GNN. The tuned hyperparameters and their ranges are the following (the best values are underlined): initial learning rate: $\{10^{-4}, \underline{10^{-3}}, 10^{-2}\}$; weight decay: $\{\underline{0}, 10^{-4}, 10^{-3}\}$; dropout: $\{0.0, \underline{0.2}\}$; number of GNN layers: $\{3, \underline{5}\}$, size of node embeddings: $\{32, 64, \underline{128}\}$; type of GNN operation: $\{\underline{\text{GCN}}, \text{GIN}, \text{GAT}\}$; readout operation: $\{\text{mean}, \underline{\text{max}}, \text{sum}\}$. Grid search hyperparameter tuning was also performed for the baselines. The best hyperparameters for them are provided in Appendix, Table 1. During hyperparameter tuning, we used validation subsets and random seeds different from those used to report our main results.

### 5.4  Performance evaluation

We use the balanced accuracy score (Brodersen et al., 2010) as our quality metric. Although we aimed at creating as balanced datasets as possible (as shown in Appendix, Figure 10), they are not perfectly balanced (multiple tail moves could still result in a network that contains the tree, though this is unlikely). Thus, we use balanced accuracy instead of standard accuracy. In all experiments, we report the results of test datasets using the epoch at which the best performance on the validation dataset was achieved.

We compare the performance of Combine-GNN with the aforementioned baselines. Furthermore, we study the scalability of Combine-GNN in terms of the training dataset size and increasing size of the test instances (in the inductive learning setup). Finally, we specifically study how the different design choices of the GNN used in our algorithm affect the performance. The results of statistical significance testing are provided in Appendix, Table 3.

We also empirically study the time-wise complexity of Combine-GNN (inference time) compared to the exact algorithm for the tree containment problem, namely, BOTCH (van Iersel et al., 2023; Huijsman, 2023). For this analysis, we specifically consider more challenging instances, excluding tree-child phylogenetic networks, because for them the tree containment can be solved in linear time. We note that, for a rigorous time analysis, we adopt a conservative approach in estimating the time performance of Combine-GNN: processing one sample at a time (without using batching) and assuming that both data and the GNN need to be transferred to the GPU each time a sample is processed. This approach differs from a more practically realistic scenario where the model is created and transferred to the GPU only once. We use a system with Intel(R) Xeon(R) Silver 4110 CPU and Nvidia GeForce RTX 2080 Ti GPUs (for this experiment, only one GPU is utilized).

## 6  Results

The main results are shown in Figure 3. We see that Combine-GNN shows good performance in both inductive and transductive learning settings: the average balanced accuracy is 0.958 and 0.970 respectively, in most runs across datasets of different sizes the accuracy is $> 0.95$. Notably, its performance does not substantially deteriorate on larger instances.

As anticipated, the baseline GNN approach, utilizing one-hot encoding of the leaves, exhibits better performance in the transductive setting (albeit substantially lower than Combine-GNN) than in the inductive setting. The baseline which uses XGBoost on the manually engineered features shows reasonably good performance, but its underperformance compared to Combine-GNN indicates that relying solely on general network and tree characteristics does not suffice to accurately determine tree containment.

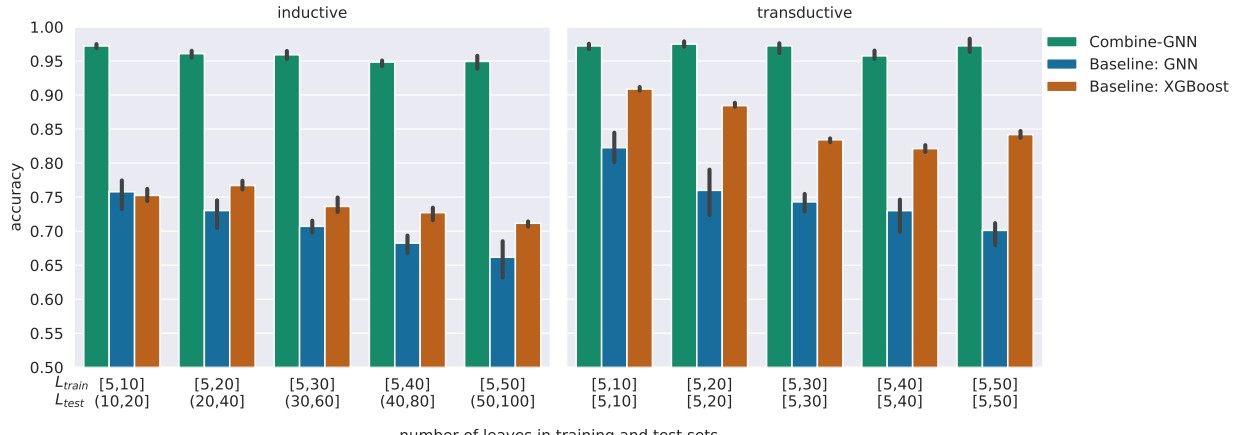

Figure 3: Main results: performance of our algorithm and the baselines in terms of (balanced) accuracy on different datasets; the left part of the graph shows the results in the inductive setting, the right part of the graph shows the results in the transductive setting. For each dataset and each algorithm, we perform five runs with different seeds. Bar height denotes the average values; error bars denote the 95% confidence interval.

## Performance on much larger instances then used for training

The performance of Combine-GNN on test datasets with instances of increasing size (including those which are much larger than the instances contained in the corresponding training dataset) is presented in Figure 4. We observe that while the performance deteriorates with the difference between size of the training and test instances increasing, Combine-GNN demonstrates rather robust behaviour, for instance, when trained on instances with up to 20 leaves, the balanced accuracy score on datasets with instances with up to 100 leaves, stays above 0.9. This highlights the generalization ability of the proposed approach.

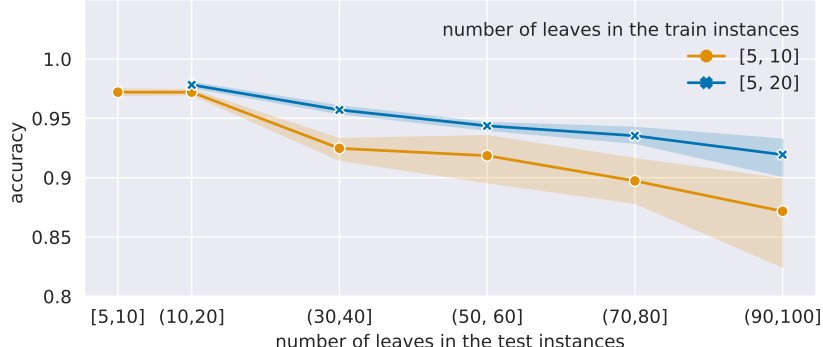

Figure 4: Performance (balanced accuracy) for test datasets containing instances with increasing number of leaves. The leftmost point in each line shows the performance in the transductive setup, the remaining ones show the performance in the inductive setup. The number of leaves in the test instances are sampled randomly uniformly from the specified interval. The results are averaged over five runs with different seeds.

## Impact of the training dataset size

The performance of Combine-GNN using training datasets of varying sizes is presented in Figure 5. For a smaller number of samples, the number of epochs has been adjusted accordingly, so that the total number of processed samples during training is equal in all cases ($10^6$). We can conclude that a larger number of samples is clearly beneficial for the performance. It is plausible that using more than 10000 samples in our

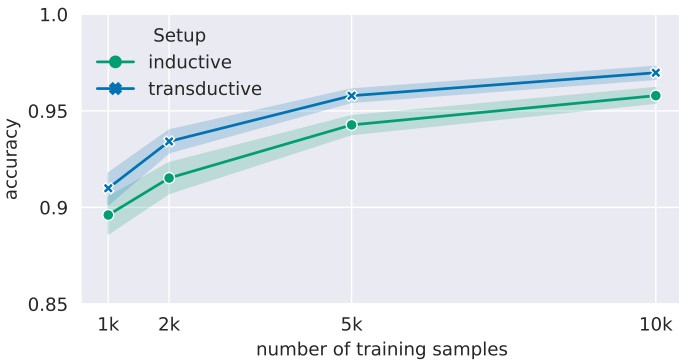

Figure 5: Performance (balanced accuracy) for training datasets of different sizes. The results are averaged over five inductive and five transductive learning setups with a different number of leaves and five runs with different seeds for each of them.

training datasets can result in higher accuracy scores. However, we did not generate larger datasets due to the computational costs of generating true labels (scaling exponentially with the instance size).

**Impact of the number of reticulation nodes**

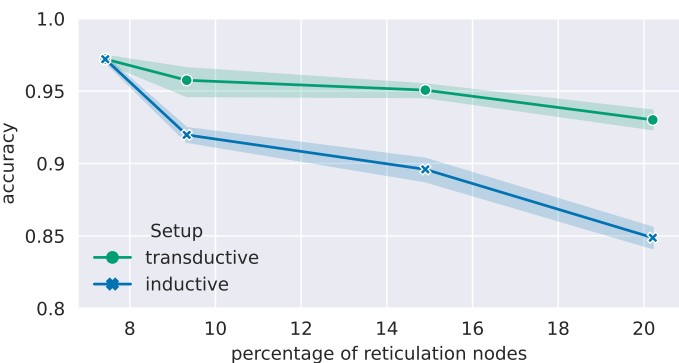

Figure 6: Performance (balanced accuracy) for training and testing datasets with varying percentage of reticulation nodes. Here, the transductive setup denotes the test instances with $L$ from $[5; 10]$ and inductive denotes the instances with $L$ from $[11; 20]$. The training instances in both setups have the $L$ from with $[5; 10]$. The results are averaged over five runs with different seeds for each setup and each percentage of reticulation nodes.

The performance of Combine-GNN with training and testing datasets of varying number of reticulation nodes is presented in Figure 6. We use as training datasets the datasets with instances with $[5; 10]$ leaves and varying number of reticulations (by controlling the parameter $\alpha$ in the LGT generator). The test datasets have been generated using the same $\alpha$ correspondingly. We can conclude that a larger percentage (number) of reticulations makes the task much more difficult for Combine-GNN to solve. However, we note that in the transductive setup, the accuracy stays above 0.9 even for the dataset with >20% reticulation nodes.

**Real-world data results**

The results for real-world data experiments are shown in Figure 7. We note the challenges posed in this experiment: the training data is a synthetic dataset generated using LGT generator (real-world data is used only as validation/test subsets); furthermore, the used hyperparameters were tuned on synthetic datasets

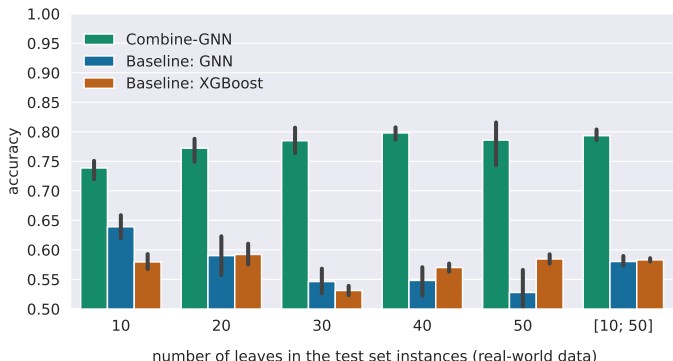

Figure 7: Performance (balanced accuracy) for real-world data. The training data was generated using LGT generator. The results are averaged over five runs with different seeds for different instance sizes.

only; the real-world validation subsets are rather small causing potential overfitting on them (we choose the best epoch based on the validation performance).

Nevertheless, under all above-mentioned challenges, Combine-GNN demonstrates reasonably good accuracy, in particular, 0.798 (averaged over five seeds) when all considered real-world data ($L \in \{10, 20, 30, 40, 50\}$), is used for validation and test subsets. It also outperforms the baselines by a substantial margin. Thus, we can conclude that Combine-GNN can generalize to the real-world data as well (in contrast to the baselines). Potentially, its performance might likely be further improved by, for instance, specific hyperparameter tuning, or using more and/or different training data.

One of the reasons for lower performance of Combine-GNN on the real-world data is the higher percentage of reticulation nodes (20% for $L = 10$, 17.6% for $L = 50$, slighly decreasing with the increase of $L$) than in the datasets used in our main experiments (Figure 3) and hyperparameter tuning (the percentage of reticulation nodes there was less than 8%). Moreover, it was shown in Bernardini et al. (2023) that the LGT generator does not generate the data that has exactly the same properties as this real-world dataset.

## 6.1 Runtime performance

The time performance (inference) results are shown in Figure 8. First, we observe that the exact algorithm (BOTCH), as anticipated, takes exponential time to solve tree containment (on instances that are not tree-child networks). Our algorithm shows polynomial time complexity, namely, the best linear (in the log-log axes) fit indicates the complexity of $\mathcal{O}(N^{0.21})$. In particular, we observe that for the considered graph sizes, the inference time itself scales extremely well (sublinearly). The data processing step (including network-tree graph creation) has approximately linear complexity, as expected. The model creation time demonstrates approximately constant time complexity (the model size does not depend on the graph size).

## 6.2 Ablation studies

The results of all ablation studies are shown in Figure 9.

### Taking into account the direction of the edges

The use of Dir-GNN demonstrates its advantage over a simpler approach that treats graphs as undirected. This is a reasonable behavior since retaining information about the edge directions in the given network and tree is expected to be helpful in solving the tree containment problem.

### Node features

We observe that adding the origin of the node (whether it belongs to the network, the tree, or is a leaf node) to the nodes of the network-tree graph as features leads to performance improvement. This is an

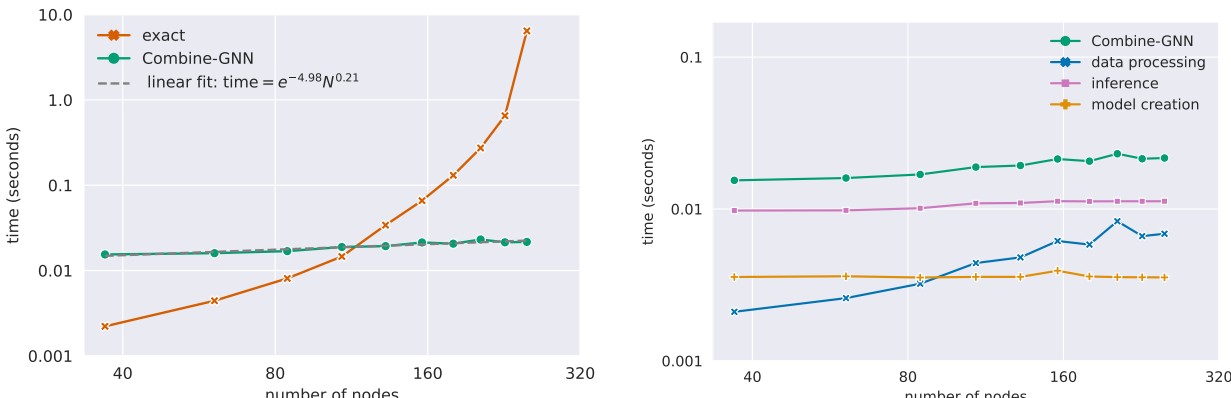

Figure 8: Left: scalability of Combine-GNN (along with the best linear fit in the log-log axes), and the exact algorithm (BOTCH) in terms of the required time to solve tree containment instances. Right: the breakdown of time consumption by Combine-GNN. We average all measurements over 10 runs, and use averaging on the x-axis over 10 bins (the points denote the centers of the bins).

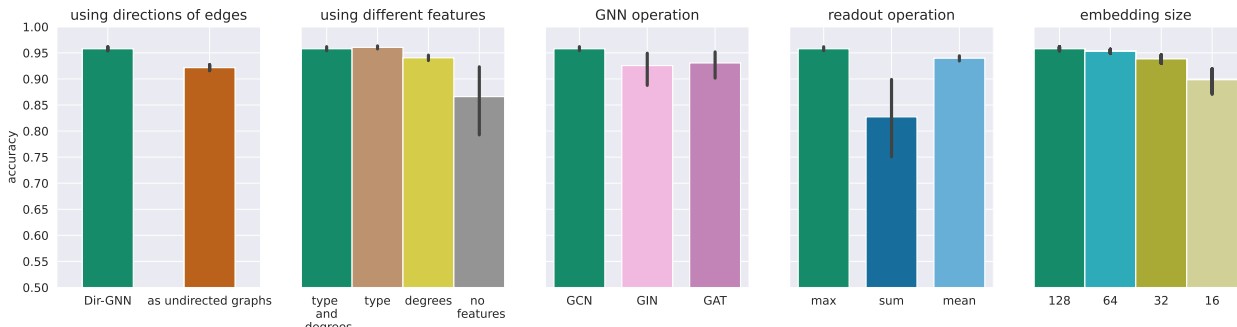

Figure 9: Effects of different design choices of Combine-GNN to the performance (balanced accuracy). Performance is averaged over five datasets in the inductive learning setup and five random seeds for each dataset (25 values in total for each shown bar). Bar height denotes the average values; error bars denote the 95% confidence interval.

anticipated result, since these supplementary features explicitly convey to the GNN whether a node pertains to the network or the tree. We note that adding the node in- and out-degrees as additional features does not increase the performance further.

**GNN architecture design choices**

We study how different GNN operations (GCN, GAT, GIN) affect the performance. We can conclude that, while the GCN operation works better than the considered alternatives, both GAT and GIN operations also demonstrate reasonable performance. This highlights that our algorithm is not strongly reliant on the GCN operation and suggests the potential for achieving even better performance by exploring different GNN operations.

Similar conclusions can be made about the graph-level aggregation operation: the used maximum pooling operation performs better than the considered alternatives (specifically, we note that the summation aggregation underperforms compared to the alternatives).

The embedding size appears to be a rather important hyperparameter: while there is no substantial difference between performance of embedding size 128 and 64, the performance substantially deteriorates with embedding size 32, and especially, 16.

## 7 Discussion

In this work, we demonstrated that GNNs can be used as approximate solvers of the tree containment problem, i.e., they can predict whether a phylogenetic network contains the given phylogenetic tree or not with high accuracy. While this capability is valuable, for practical applications, an algorithm that not only provides containment predictions but also offers a solution representation might be advantageous. For instance, such an algorithm could generate a mapping between the nodes of the tree and the network, enabling solution verification. This verification process might help in filtering out false positives in determining tree containment: ensuring that a positive answer consistently reflects the true containment, although not all true positive cases may be correctly identified as such. We consider this direction as practically important direction of future research.

Another important direction of future work is exploring whether the proposed approach (or adaptations of it) can be applied to solving more general formulations of the tree containment problem, e.g., non-binary phylogenetic networks and the network (instead of a tree) containment. Such an adaptation might ease the application of the proposed approach to real-world phylogenetic datasets.

There are other problems in phylogenetics related to TREE CONTAINMENT, for instance, HYBRIDIZATION. For that problem, the input consists of multiple trees and the aim is to construct a phylogenetic network with at most $k$ reticulation nodes such that it contains all input trees. Potentially, at least some of the ideas proposed in this work can be used to solve other phylogenetic problems such as HYBRIDIZATION, though we acknowledge that this a non-trivial task and requires additional research.

## 8 Conclusion

In this work, we demonstrated how GNNs can be used to approximately solve an important NP-complete problem from the phylogenetics field: TREE CONTAINMENT. We proposed an approach named *Combine-GNN* that allows inductive learning: it consists of combining the given phylogenetic network and tree in a single graph and applying a GNN to it. This way, we achieve the inductive learning ability: it can work for instances with a larger number of leaves (i.e., studied species) than the number of leaves of the training instances. Combine-GNN demonstrates high accuracy in both transductive and inductive learning setups with instances having up to 100 leaves, clearly outperforming more simple baseline approaches. Furthermore, we carefully studied how different design choices of our algorithm affect its performance. Overall, our results demonstrate that GNNs have grate potential in phylogenetics.

### Acknowledgements

The work in this paper is supported by the Dutch Research Council (NWO) through project OCENW.GROOT.2019.015 "Optimization for and with Machine Learning (OPTIMAL)".

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

## Appendix

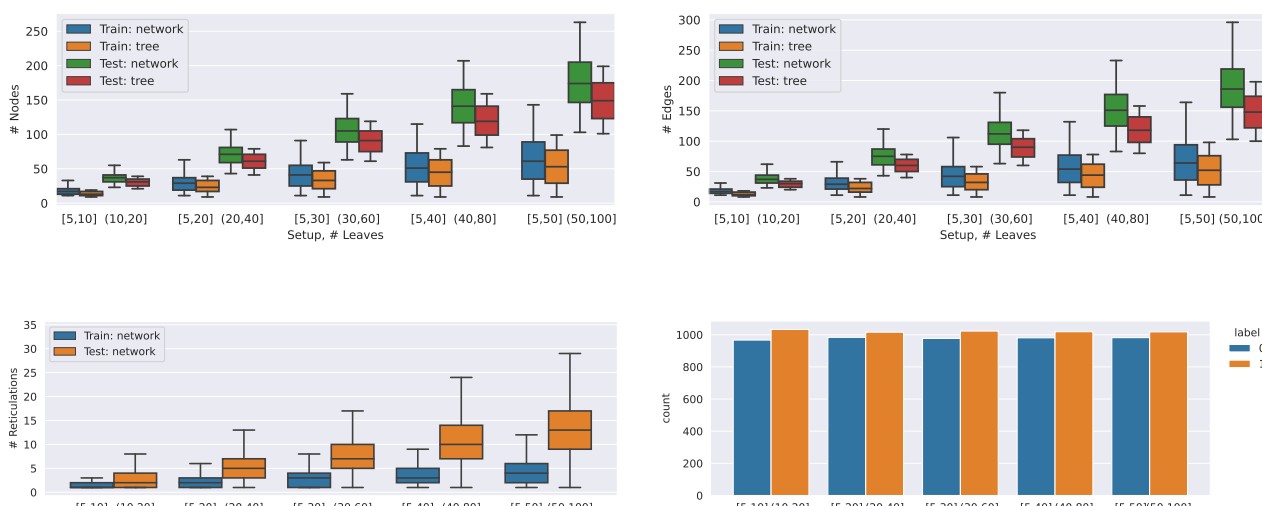

Figure 10: Number of nodes (upper left), number of edges (upper right), number of reticulation nodes (bottom left) and label (whether the tree is contained or not) statistics of the graphs in the used datasets.

| GNN-based baseline | |
|---|---|
| Hyperparameter | Considered values |
| initial learning rate | $\{10^{-4}, \underline{10^{-3}}, 10^{-2}\}$ |
| weight decay | $\{0, \underline{10^{-4}}, 10^{-3}\}$ |
| dropout | $\{\underline{0.0}, 0.2\}$ |
| number of GNN layers | $\{3, \underline{5}\}$ |
| size of node embeddings | $\{32, 64, \underline{128}\}$ |
| type of GNN operation | $\{$GCN, $\underline{\text{GIN}}$, GAT$\}$ |
| readout operation | $\{\underline{\text{mean}}, \text{max}, \text{sum}\}$ |

| XGBoost-based baseline | |
|---|---|
| Hyperparameter | Considered values |
| estimators | $\{\underline{50}, 300\}$ |
| max depth | $\{3, 5, 10, \underline{\text{None}}\}$ |
| learning rate | $\{10^{-4}, 10^{-3}, 10^{-2}, \underline{10^{-1}}\}$ |
| max number of leaves | $\{0, 10, \underline{100}\}$ |
| subsample | $\{0.5, \underline{0.9}\}$ |

Table 1: Search spaces of hyperparameters for the considered baselines. The best values are underlined.

| basic calculated values | calculated features |
|---|---|
| $\#N, \#T$: number of nodes in the network and the tree respectively | $\#T/\#N$ |
| $N_{depth}$ - network depth, $T_{depth}$ - tree depth | $N_{depth}/\#N$ 
 $T_{depth}/\#T$ |
| $d_{network}, d_{tree}$ : arrays of distances to the corresponding leaves in network and tree respectively | $\min(d_{network})/N_{depth}$ 
 $\text{mean}(d_{network})/N_{depth}$ 
 $\min(d_{tree})/T_{depth}$ 
 $\text{mean}(d_{tree})/T_{depth}$ 
 $\min(d_{network}) - \max(d_{tree})$ 
 $(\min(d_{network}) - \max(d_{tree}))/\#N$ 
 $\min(d_{network} - d_{tree})$ 
 $\max(d_{network} - d_{tree})$ 
 $\text{mean}(d_{network} - d_{tree})$ |

Table 2: The features used in the XGBoost-based baseline.

| Experiment | Hypothesis | p-value |
|---|---|---|
| Main results | Combine-GNN performs better than GNN baseline
Combine-GNN performs better than XGBoost baseline | **0.000**
**0.000** |
| Ablation: GNN operation | GCN performs better than GIN
GCN performs better than GAT | **0.011**
**0.01** |
| Ablation: readout operation | max performs better than sum
max performs better than mean | **0.001**
**0.000** |
| Ablation: embedding size | embedding size 128 performs better than embedding size 64
embedding size 128 performs better than embedding size 32
embedding size 128 performs better than embedding size 16 | 0.063
**0.000**
**0.000** |
| Ablation: using graph directionality | Dir-GNN performs better standard GNN (on undirected graphs) | **0.000** |
| Ablation: using different features | Using node type features and node degree features performs better than using node type features
Using node type features and node degree features performs better than using degree features
Using node type features and node degree features performs better than using no features | 0.336
**0.000**
**0.000** |

Table 3: The results of statistical significance tests. We use the one-sided Wilcoxon test, testing differences between the balanced accuracy scores. Statistically significant results at $\alpha = 0.05$ and applied Bonferroni correction (in each experiment, we apply the correction with $n$ set to the number of tested hypotheses) are highlighted with bold font. The p-values are rounded to three decimal places.

| Setup | Balanced accuracy | Precision | Recall |
|---|---|---|---|
| transductive | 0.970 | 0.958 | 0.986 |
| inductive | 0.957 | 0.942 | 0.976 |

Table 4: Balanced accuracy along with precision and recall metrics for Combine-GNN. These numbers are averaged over all inductive and transductive setups from Figure 3 (main results).

