# OpenReview forum: "Solving the Tree Containment Problem Using Graph Neural Networks"
_TMLR — Accepted by TMLR_

### Review · Reviewer_zcx7 · 2024-03-10

**Summary Of Contributions:**

The paper addresses the tree containment problem for binary phylogenetic networks and provides the first approximate inference algorithm based on a graph neural network. A key insight of the proposed technique is that the construction of a “display graph”, by merging the tree and the network into a single graph, making the leaf nodes common, is a vital step in rendering the GNN effective. The paper provides the results of an experimental analysis using synthetic data. These results indicate that the approach significantly outperforms a naïve GNN method or a xgboost method trained on handcrafted features. Results of ablation studies are provided to demonstrate the importance of different design choices.

**Audience:**

Yes

**Broader Impact Concerns:**

No concerns

**Claims And Evidence:**

No

**Requested Changes:**

1.	The paper does not adequately motivate the use of a machine learning algorithm for this application. In particular, there is no explanation of how an approximate solution is useful. If the task is to determine “whether a given phylogenetic tree is contained…”,  under which circumstances is it useful/acceptable to provide an answer that is only correct 95 percent of the time? Please provide a clearer explanation of how the approximate algorithm can be used satisfactorily in practice. Does the false-alarm rate matter? Is accuracy the most appropriate metric?

2.	The related work section on GNNs should be expanded to include more recent works. Given the enormous amount of work over the past 4 years, it is not reasonable to cite papers from 2017-2019. Yes, these are important papers that should be cited, but there have been major advances since then. By citing only a single paper (Rossi et al., 2023), the paper gives a misleading impression of the field. Requested change: Since the development of a novel GNN is not the focus of the paper, it would suffice to cite a recent review paper and include a couple of sentences that acknowledge that MPNNs are not the only approach.

3.	The sentence “Through this technique, we implicitly incorporate the labels of the leaves without directly using specific leaves’ labels. This approach holds the potential to enable inductive learning ability since the labels of specific leaves are not utilized in any form.”
These two sentences are difficult to understand. They appear contradictory and it is not clear what support there is for the claim in the second sentence. The first sentence states that “we implicitly incorporate the labels”. The second sentence states that “the labels of specific leaves are not utilized in any form”. If they are not utilized (in any form), then why and how are they implicitly incorporated? Or is the word “specific” supposed to be important?

The second sentence is a relatively strong claim and it is not supported by any argument or citation. Perhaps the claim should be reversed? There is a different meaning to the sentence “By avoiding the explicit use of the labels of specific leaves, this approach does not prevent inductive learning from occurring”.

4.	I don’t understand the purpose of presenting the application of the undirected GNN in Section 4.1 if the following Section states that the method actually uses a directed GNN. It would be much clearer just to present the directed GNN directly. If the authors feel that is important to present the operation of an undirected GNN as background material, then it would be better to include it in a separate section. I think the TMLR readership has sufficient familiarity with GNNs that there is no need. Readers who are unfamiliar can consult the paper by Gilmer or Hamilton’s book, for example. Requested change: Please move Section 4.2 into Section 4.1 so that the summary of the technique is consistent with what is actually implemented.

5.	The approach appears heuristic in nature, amounting to “we construct a display graph and apply a directed GNN”. Can the authors provide a stronger argument as to why this approach works? Even if it is not possible to provide theory, the paper would be much stronger if it could provide an explanation of how the GNN can learn whether an injective mapping exists or not. Please provide some additional explanation or analysis that provides considerably greater insight into what is being learned.

6.	Please explain why the paper only considers binary phylogenetic networks. It is not obvious that there is any aspect of the methodology that prevents application beyond the binary case.

7.	The proposed technique is only tested on one synthetic dataset. Presumably the intention of developing this approach is not to apply the technique to synthetic data. Where is the evidence that this synthetic generation process leads to data that has the same distribution as real data? Why is it not possible to provide some example experimental results for real data?

8.	Please provide more discussion and analysis of the dataset. Is it clear/known that the tail operations do not lead to an obviously different structure (in terms of basic graph properties)?

9.	The discussion of the ablation study is insufficiently detailed. It is not enough to state that “summation aggregation underperforms” without any explanation. Why does it underperform? Summation is supposed to exhibit stronger inference capability than an averaging operation (see, e.g., the GIN paper). A result like the one presented here is concerning and needs to be explained. Please explain why summation underperforms and justify the explanation.

10.	Along the same lines, it is not enough to state that “GAT and GIN operations also demonstrate reasonable performance”. Is GCN’s performance difference statistically significant? If so, why is it the case? Usually, we expect GIN to be superior when it comes to learning structural properties of a graph. A GCN is usually able to process feature-information better. Does this imply that the feature information is more important for this problem? The ablation that removes the node origin suggests that this is not the case. Please explain why GCN achieves a better performance and justify the explanation.

11.	Please add the results (or at least a summary) of a similar ablation study for removing the node degree feature.

12.	Please confirm that the degree distributions for the positive and negative samples in the dataset are indistinguishable. Does the xgboost baseline (or a simple MLP) perform better if it is trained using the degree distributions?

13.	Please explain why simple graph features are not used in the xgboost baseline? For a graph with hundreds of nodes, it is relatively straightforward to determine numerous basic graph statistics.

14.	Please conduct statistical significance tests where appropriate.

Minor

15.	The text on the right-hand-side of Figure 2 is far too small. Please increase the font size. The text in the dark-coloured boxes is very hard to read. Please change the colour or move the text.

16.	Not mandatory, but suggested: In Figure 1, it is not obvious that leaf nodes A and B have been re-ordered in (c). The leaf node labels are small – perhaps the reordering can be highlighted by adding colours or larger labels.

17.	The font size in Figure 8 is far too small.

**Strengths And Weaknesses:**

Strengths

1.	The paper is the first to study the binary containment problem using a graph neural network approach.

2.	The application of a directed GNN to the constructed display graph is innovative and effective.

3.	The paper is clearly written. The binary containment problem is clearly explained for a non-expert (which is likely the case for most of the TMLR readership).

Weaknesses

1.	Although the approach is innovative and appears effective, the technical contribution is
relatively minor. It amounts to constructing a display graph (which the authors state is common in the phylogenetics literature) and then applying a directed GNN (without any modification of the algorithm). The paper does not provide any insightful discussion or experimental analysis to explain why this approach is effective.

2.	The paper does not provide a motivating text that explains why an approximate inference algorithm is satisfactory and how it would be used in practice. With this in mind, there is no clear justification of the selected performance metric.

3.	The experiments are conducted for a single synthetic dataset. The paper does not provide any evidence that the proposed generation mechanism leads to networks that display the same important properties as the networks that would be encountered in a practical setting. With this in mind, it is not clear whether the proposed algorithm is only exploiting some distributional aspect of the generative mechanism that would disappear in a real dataset.

4.	The ablation analysis does not contain enough detail. There are concerning experimental outcomes (outperformance of GCN, underperformance of summation). Given that there are no node features, the intuition is that the structural aspects of the constructed graph should be most important. Under such circumstances, an algorithmic combination such as GIN and summation would be expected to be the best choice. The paper should explain why this is not the case. Without such an analysis, there is a concern that the algorithm is learning some artefact of the synthetic data generation process.

---

> ### Author Response · Authors · 2024-04-01
>
> We thank the reviewer for the feedback.
> We have done our best to revise the paper according to the requested changes:
>
> 1. We have added an explanation of how an ML approach can be used to the Introduction section (at the end of the first contribution paragraph): “Our approach can be applied to evaluate the quality of a constructed phylogenetic network: it can be used to efficiently estimate whether trees (e.g., gene trees) are contained in the built network with high accuracy”. While we believe balanced accuracy is a suitable metric to evaluate and demonstrate the general performance of our method, we additionally calculated precision and recall metrics (appendix, table 4).
> We observe high recall (0.976) and relatively slightly lower precision (0.942) (but also high in absolute value). Since we do not observe abnormally lower values for either precision or recall, and they both might be relevant in practical usage (we might not want to wrongly discard good networks, but also to wrongly accept bad ones), we stick to the balanced accuracy as our main performance evaluation metric.
> Furthermore, we believe this work in general shows the potential of GNNs in solving phylogenetic problems, and might motivate different applications of GNNs to other phylogenetic problems. We also add this as the 3rd contribution of our paper.
>
> 2. We have adjusted the related literature section according to the reviewer’s suggestion, adding a recent survey paper. However, we have decided to keep the references to such fundamental papers as GCN and GIN, though they’re not very recent.
>
> 3. We have rephrased and added more details to these sentences. Our main message here is that, for our approach, at least theoretically, it is possible to work in the inductive setting, since the leaves’ labels outside the training set do not pose a problem in constructing the network-tree graph (in contrast to using the labels as node features). We believe this claim is not too strong, since we use the word “potentially” and leave the actual (empirical) verification of the claim to be seen in the Results section.
>
> 4. We agree with the reviewer’s proposal and have changed the 4.1-4.2 sections accordingly
>
> 5. We agree that our approach not only appears to be heuristic in nature, it is in fact an heuristic. The choice for GNNs to tackle our tree containment problem was given in by the fact that other hard combinatorial graph problem have been tackled by GNNs successfully
>  (but also only heuristically) as we extensively describe in Section 2, under Solving hard problems with Graph Neural Networks.
>  We admit that we do not have any convincing insights or theory that we could present on what are the best algorithmic choices for given combinatorial problems. To the best of our knowledge, there are no other papers claiming to have such insights.
>  The only thing we can conclude is that empirical experiments in our paper show that our intuition for this problem was pretty good.
>
> 6. Indeed, our approach can be theoretically applied to non-binary networks as well. However, in this paper, we consider the binary case only, mainly because in the phylogenetic literature tree containment has been so far considered for binary networks only. From the practical point of view, adding the non-binary case is difficult, because we need an exact tree containment algorithm to determine the ground truth labels and there are no fast algorithms that can be used to generate our training data.
>
> 7. We have added the results on real-world data to the revised version of the paper (average accuracy is ~0.8 with training data being synthetic only, the details are provided in the paper revision).
> We also note that in the used synthetically generated data, the generated networks are not random graphs: we used the Lateral Gene Transfer (LGT) generator from Pons et al. (2019), where it was shown that it represents a viable evolutionary model, possible to occur in nature (but the used real-world data instances are not exactly the same as from the LGT generator as shown in  Bernardini et al. (2023)).
>
> 8. Tail moves just relocate a node from one edge to another. Thus, the main graph properties (#nodes, #edges, etc.) are not changed. We have added a clarifying sentence to Section 5.1
>
> 9. Our hypothesis about the reason why max pooling graph-level aggregation (and average aggregation) outperforms summation is that the summation result is strongly connected with the display graph size (#nodes), and therefore works not so well especially in the inductive setting. In the tree containment problem, the display graph size does not help to get the correct prediction on its own. In our experiments, we observe that the sum aggregation performs especially bad on bigger instances (starting from graphs with up to 60 leaves).

---

> ### Author Response · Authors · 2024-04-01
>
> 10. First, we’d like to argue that we use not GCN and GIN directly, but instead their directed generalizations inspired by the Dir-GNN paper. Furthermore, GIN wasn’t part of Dir-GNN paper, but Dir-GCN outperformed Dir-SAGE and Dir-GAT there. We also note that on a somewhat similar graph matching problem (Lou, Zhaoyu, et al. "Neural subgraph matching." arXiv preprint arXiv:2007.03092 (2020)), GIN is outperformed by GCN as well as by SAGE. Finally, we note that in our ablation studies, we fix all hyperparameters except for the ablated one, and theoretically, a different configuration might benefit GIN more. Nevertheless, we believe the difference between GCN, GIN, and GAT is not dramatic, and, this highlights that our Combine-GNN is not strongly dependent on one particular GNN architecture.
>
> 11. We have added this requested ablation study.
>
> 12. That’s correct, the degree distributions are indistinguishable, since the tail moves do not change them (they just move a node from one edge to another). That’s also the reason why we do not use the degree distribution in some form to train xgboost models.
>
> 13. The answer is similar to the previous one: since tail moves do not change the node degrees, the number of edges, the number of leaves, etc., we focus on the features which, in our opinion, are more relevant to the graph containment: for instance, the features concerning the labels of the corresponding leaves (in the tree and the network).
>
> 14. We have added (to the Appendix) the results of statistical testing of the performance differences between algorithms and Combine-GNN design choices. Most of the differences turn out to be statistically significant (at alpha 0.05).
>
> 15. We have enlarged the font size and tried to make it easier to read
>
> 16. To not overwhelm the picture with different colors, we have added a textual note about the labels order.
>
> 17. We have enlarged the font size on this Figure.

---

### Review · Reviewer_Yakq · 2024-03-14

**Summary Of Contributions:**

The authors propose to approximately solve the NP-complete problem of tree containment using a graph neural network approach. They empirically show excellent predictive performance, even in an inductive setting.

**Audience:**

Yes

**Claims And Evidence:**

No

**Requested Changes:**

The dataset is constructed generating random binary trees with 5 to 100 leaves (and up to 200 nodes in total) and generating 10k examples for training and 1k examples for testing. With this amount of training the predictive performance is ~ 95% in the inductive setting, i.e. when training is done on graphs that are half of the size of the graphs used in test.

I wonder if one could try to asses if the trees in the test set are not subgraphs of the trees in the training set. Wouldn't that constitute a form of data leak between the train and test if that would occur?

It would be informative to see some notion of performance degradation profile:
- a learning curve would show the performance when increasingly less material is used for training
- a graph that shows the performance vs the increase of the size ratio between the training and the test set should also be of interest (in the experiments reported the ratio is always 1:2)
- increasing the fraction of reticulations w.r.t. total nodes would be of interest as this affects the complexity of the problem to solve (in the experiments the ratio seems also fixed to ~ 1:10)
- showing the relation between the predictive performance and the size of the node embeddings would also be informative

**Strengths And Weaknesses:**

Pos:
- the idea of combining the needle (tree) and the haystack (the network) so not to have to provide an explicit identity to the nodes is interesting
- the ablation study shows which elements are important in the solution (as it turns out, the only important choice is to use a max or mean readout rather than a sum).

Neg:
- (acknowledged by the authors) the approach does not provide a mapping between the nodes of the tree and the nodes of the network so no verification can easily be carried out, i.e. the method is a black box oracle
- no explanation is offered into understanding what can possibly have the model learned to be so successful in its predictions: the presence of reticulation nodes (up to 15 in a 200 nodes network) should exponentially mix the messages coming from the nodes, and the model would have to propagate information from (distant) tree nodes (without suffering from squashing effects) to learn to which nodes to pay attention to when aggregating information (although GATs are shown to be actually  performing worst than GCNs in this problem!). Without such knowledge it's hard to understand the limitations of the approach, i.e. have an intuition on when it will fail, or how it will degrade.

---

> ### Author Response · Authors · 2024-04-01
>
> We thank the reviewer for the feedback.
> We have done our best to revise the paper according to the requested changes:
>
> - We believe there is no data leak in our datasets generation: the answer to the tree containment problem depends on the network also, and in the inductive setup, the networks in the validation set have more leaves than networks in the training set.
>
> - We believe Figure 5  in the paper sufficiently answers this question: we see that increasing the training dataset size from 1000 to 2000, and from 2000 to 5000 give substantial improvements, from 5000 to 10000 the improvement becomes smaller though still exists.
>
> - We believe Figure 4 answers this: as expected the accuracy is decreasing while the difference between testing and training instances sizes is increasing. When trained on instances with [5;10] leaves, the accuracy on instances with (50;100] leaves, drops below 0.9. However, when trained on instances with [5;20] leaves, the accuracy on instances with (50;100] leaves, stays above 0.9, but also visibly drops.
>
> - We have added experiments on the instances with more (larger percentage of) reticulation nodes. As expected, more reticulations make the task harder for Combine-GNN too, however, the accuracy stays above 0.9 in the transductive setup and above 0.85 in the inductive setup.
>
> - We have added the results with different embedding sizes: “The embedding size appears to be an important hyperparameter: while there is no substantial difference between performance of embedding size 128 and 64, the performance substantially deteriorates with embedding size 32, and especially, 16.”

---

### Review · Reviewer_FHbL · 2024-03-21

**Summary Of Contributions:**

The paper proposes a new approach to dealing with the tree containment problem in phylogenetics. It combines network and tree to generate a graph first, and then uses GNNs for prediction. The resultant model achieves a high accuracy of over 95% for instances with up to 100 leaves. Its performance is much better than baseline methods.

**Audience:**

Yes

**Claims And Evidence:**

No

**Requested Changes:**

• In Section 4, it is claimed that “implicitly incorporate the labels of the leaves without directly using specific leaves’ labels. This approach holds the potential to enable inductive learning ability since the labels of specific leaves are not utilized in any form.” Besides the empirical results, it would be better to provide theoretic analysis on generalization ability to support this claim.
• Only using GNN and XGBoost as baseline methods for comparison is not sufficient for comparison. Please include more SOTA methods.
• In Section 6.2, it seems that some experimental results for ablation studies are close to each other. Please also provide standard deviation in addition to accuracy.
• Based on the experimental results, the performance of the proposed method is much better than that of baselines. Is there any limitation of this method? Please discuss it.

**Strengths And Weaknesses:**

Strengths
• Applying Graph Neural Networks to solve the tree containment problem in phylogenetics is novel.
• Based on the experimental results, the proposed approach shows a good generalization ability and achieves high accuracy, even though the training dataset is small.

Weaknesses
• The introduction of the background and the problem is brief. As a result, it is less accessible to people who are not in this specific field.
• The authors need to provide more technical analysis and more comparisons in experiments to convince readers.

---

> ### Author Response · Authors · 2024-04-01
>
> We thank the reviewer for the feedback.
> We have done our best to revise the paper according to the requested changes:
>
> - In this version of the paper, we rephrased and elaborated on this claim. Our main message here is that, for our approach, at least theoretically, it is possible to work in the inductive setting since the leaves’ labels outside the training set do not pose a problem in constructing the network-tree graph (in contrast to using the labels explicitly as node features). We believe this claim is not too strong, since we use the word “potentially” and leave the actual (empirical) verification of the claim to be seen in the Results section.
>
> - To the best of our knowledge, in this work we apply a machine learning-based approach to approximately solving the Tree Containment problem for the first time. Thus, there are no other established baselines (and no commonly known SOTA method) for approximate Tree Containment solving. However, we show the time-wise comparison with the commonly used exact solver as well.
>
> - The error bars on the plots show the 95% confidence intervals, additionally in this version of the paper we added the statistical tests. Most of the performance differences turn out to be statistically significant at alpha=0.05.
>
> - We believe we sufficiently address the limitations of our method in the discussion: main limitation is the black box nature of it (for instance, in contrast to the xgboost baseline on the manually constructed features).

---

### Author Response · Authors · 2024-04-01
**General reply to the reviews**

We thank the reviewers for their detailed feedback. We have addressed the requested changes in the new version of the paper and elaborated on them in detail in our replies to each reviewer.
Here we’d like to address the mentioned general weaknesses of the paper as they were mostly shared by all reviews:

*The black-box nature of the proposed approach*:
    we agree that our method does not provide explanations of the answer it is giving to the tree containment problem (this is acknowledged in the Discussion). First, we'd like to argue that our approach is in the spirit of deep learning for graphs field: our design allows an end-to-end differentiable learning of relevant node and graph representations (including the inductive learning scenario). There are examples, e.g., [Ranjan, Rishabh, et al. "GREED: A neural framework for learning graph distance functions." Advances in Neural Information Processing Systems 35 (2022): 22518-22530.] when an end-to-end black-box learning approach outperforms white-box alternatives.

Then, we would like to argue that it might be extremely challenging to obtain the explanations of the model GNN predictions in human-accessible form. Such a task is in the scope of ML explainability field and we believe this is well beyond the scope of this paper and requires a lot of additional research effort. Instead of directly explaining the model predictions, we believe the massive empirical evidence and ablation studies convincingly show the usefulness of our approach and practical potential. Finally, we note that we open-source the code, so that the audience can analyze it, apply it to different datasets, and even use some of its components in applications to different problems.

*Lack of empirical evidence and variety of the considered datasets*: we believe we have improved our paper in this regard by adding new ablation studies, and experiments on new datasets (synthetic data with more reticulation nodes, and real-world data).

*No detailed explanation of the problem applications and applications of the proposed approximate solver specifically*: we have added more details on this topic. In the new version of the paper, the list of contributions of our work is expanded to include the description of potential applications and usage of our method.

---

### Decision · Action_Editor_uVX6 · 2024-05-28

**Recommendation:** Accept as is

**Comment:**

As described above, the submission's claims are supported by accurate, convincing, and clear evidence. Moreover, at least some individuals in TMLR's audience would be interested in knowing the findings of this paper.

**Audience:**

TMLR readers who already work in phyogenetics or related fields, and others from the graph learning community interested in learning about potential new applications, may be interested in this paper.

**Claims And Evidence:**

This paper studies the tree containment problem for phylogenetic networks. The authors use a GNN to estimate the likelihood that a given tree is contained within a given network. The paper’s claim that a GNN is suitable for this task is convincing given the empirical evidence, especially given the additional experiments the authors performed following the reviews.

However, I highly recommend the authors further expand on the motivation for this problem since the approach does not provide a mapping between the nodes of the tree and the nodes of the network. The authors did add an additional sentence about this point in the revision, but the paper would certainly benefit from more motivation.